# A comparison of aquaporin expression in mosquito larvae (*Aedes aegypti*) that develop in hypo-osmotic freshwater and iso-osmotic brackish water

**Lidiya Misyura**[ID]**, Elia Grieco Guardian, Andrea Claire Durant, Andrew Donini**[ID]*****

Department of Biology, York University, Toronto, Ontario, Canada

* adonini@yorku.ca

**Data Availability Statement:** All quantitative data is included in this submission in two separate spreadsheet files with all analysis equations

## Abstract

The mosquito *Aedes aegypti* vectors the arboviral diseases yellow fever, dengue, Zika and chikungunya. Larvae are usually found developing in freshwater; however, more recently they have been increasingly found in brackish water, potential habitats which are traditionally ignored by mosquito control programs. *Aedes aegypti* larvae are osmo-regulators maintaining their hemolymph osmolarity in a range of ~ 250 to 300 mOsmol $l^{-1}$. In freshwater, the larvae must excrete excess water while conserving ions while in brackish water, they must alleviate an accumulation of salts. The compensatory physiological mechanisms must involve the transport of ions and water but little is known about the water transport mechanisms in the osmoregulatory organs of these larvae. Water traverses cellular membranes predominantly through transmembrane proteins named aquaporins (AQPs) and *Aedes aegypti* possesses 6 AQP homologues (AaAQP1 to 6). The objective of this study was to determine if larvae that develop in freshwater or brackish water have differential aquaporin expression in osmoregulatory organs, which could inform us about the relative importance and function of aquaporins to mosquito survival under these different osmotic conditions. We found that AaAQP transcript abundance was similar in organs of freshwater and brackish water mosquito larvae. Furthermore, in the Malpighian tubules and hindgut AaAQP protein abundance was unaffected by the rearing conditions, but in the gastric caeca the protein level of one aquaporin, AaAQP1 was elevated in brackish water. We found that AaAQP1 was expressed apically while AaAQP4 and AaAQP5 were found to be apical and/ or basal in the epithelia of osmoregulatory organs. Overall, the results suggest that aquaporin expression in the osmoregulatory organs is mostly consistent between larvae that are developing in freshwater and brackish water. This suggests that aquaporins may not have major roles in adapting to longterm survival in brackish water or that aquaporin function may be regulated by other mechanisms like post-translational modifications.

included. All statistical results are provided in a third spreadsheet file. All these have been uploaded as Supporting Information.

**Funding:** This work was supported by a Natural Sciences and Engineering Research Council of Canada (NSERC) Discovery Grant to AD Grant #: RGPIN-2018-05841. EGG was supported by a NSERC Canada Graduate Scholarship-Masters and ACD was supported by an Ontario Graduate Scholarship.

**Competing interests:** The authors have declared that no competing interests exist.

# Introduction

*Aedes aegypti* mosquitoes are arboviral vectors for human diseases including Zika, chikungunya, dengue, and yellow fever [1]. The aquatic larvae develop in stagnant bodies of water including ponds, puddles, temporary water filled containers, marshes and swamps where the composition of this water can vary greatly, imposing different osmoregulatory challenges [2, 3]. Typically, larvae of *A. aegypti* are found in freshwater (FW) but are capable of developing in salt water with an osmolarity up to ~300 mosmol $l^{-1}$ which is iso-osmotic with their hemolymph and which will be referred to as brackish water (BW) in this study [3–6]. The salinization of FW resulting from anthropogenic activities and a climate driven rise in sea level is a global concern [7–9]. Understanding the underlying osmoregulatory physiology of mosquito larvae in these different osmotic conditions can inform public health on the potential spread of *A. aegypti*, as well as other mosquito species that live in temperate regions facing salination of habitats from anthropogenic activities.

When larvae develop in typical FW conditions, excess water must be excreted while conserving salts, whereas in brackish water (BW), larvae must counteract an accumulation of salts. Regulation of salt and water balance is accomplished by the integrated functions of organs like the gastric caeca (GC), hindgut (HG), Malpighian tubules (MTs) and anal papillae [10]. The anal papillae are external organs in direct contact with the water and in freshwater they are responsible for uptake of ions and water [11–15]. Studies have demonstrated that changes in the structure and function of anal papillae of *A. aegypti* larvae from FW adapted populations occur when these larvae are transferred to BW [16–18]. These alterations suggest a reduction in ion uptake activity when larvae are placed in BW; however, alterations of anal papillae size in response to BW depend on whether larvae come from a population adapted to FW or BW [18]. Differences in aquaporin (water channel) expression have also been documented in the anal papillae of *A. aegypti* larvae that develop in BW compared with those in FW but the functional significance remains to be studied [19]. Information on aquaporin expression and function in other osmoregulatory organs of mosquito larvae in response to BW is lacking.

Larvae ingest water from their habitat where it is directed to the GC [20, 21]. Studies have demonstrated that the GC express ion motive pumps and are sites of ion transport, and that ion transport rates are affected by serotonin, a neuroendocrine factor [21–23]. When *A. aegypti* larvae develop in BW there is a rearrangement in the localization of the ion motive pumps and there are differences in the rates of ion transport and effects of serotonin in the GC compared with the FW larvae [20, 21]. It was proposed that relatively higher rates of ion secretion from the hemolymph into the GC lumen of the FW larvae favours the secretion of water in the same direction [21]. This would provide a way to eliminate excess water from the hemolymph that enters the body as a consequence of the dilute habitat [21]. The expression and localization of aquaporins is required to fully understand the function of the GC and how these parameters are affected by development of larvae in FW and BW.

The MTs and HG are the principle organs that regulate water levels in body fluids of larvae inhabiting FW [10]. The MTs are composed of two cell types, a large principal cell and relatively small stellate cell which intercalates between the principal cells in the distal portions of the MTs. Active transport of salts from the hemolymph into the lumen of the MTs creates an osmotic gradient which drives water into the MTs. The salts are then reabsorbed back into the hemolymph by the HG and the excess water is excreted. Aquaporin transcripts have been detected in the MTs and HG of *A. aegypti* larvae [15]. Protein expression, localization and the water transport function of an aquaporin (AaAQP5) in the MTs of *A. aegypti* larvae reared in FW has also been described [24]; however, it is unknown how BW development impacts aquaporin expression in MTs and HG of larvae.

The insect aquaporins have been classified into five groups: *Drosophila* integral proteins (DRIP), big brain (BIB), *Pyrocoelia rufa* integral protein (PRIP), entomoglyceroporins and unorthodox AQP12-like [25, 26]. Six aquaporin homologs have been identified in *Aedes aegypti*, AaAQP1–6, following the nomenclature proposed by Drake et al. [27, 28]. The first AQP to be cloned from *A. aegypti* was AaAQP1, which shares highest sequence homology with DRIP and was demonstrated to preferentially transport water [27, 29]. AaAQP2 is homologous to the PRIP AQP which possesses high water conductance based on studies in a heterologous oocyte expression system [30]. AaAQP3 is homologous to the *Drosophila* BIB [30]. AaAQP4 is an entomoglyceroporin and was shown to possess higher affinity for polyols than water, as well as displaying affinity for trehalose in a heterologous expression system [30]. AaAQP5 is also an entomoglyceroporin capable of transporting trehalose and water, but not glycerol [30]. AaAQP6 is closely related to *Drosophila* CG12251, vertebrate AQP11 and AQP12 and is characterized as an unorthodox AQP and there is evidence that it transports water [26].

In this study we were interested in determining if aquaporin expression is regulated in the GC, MTs and HG, similar to what has been observed in the anal papillae [19], depending on the salinity of the water in which the larvae develop. To this end, a comparison of the transcript abundance of each of the AaAQP genes in these organs of larvae reared in FW was compared with those of larvae reared from egg in BW. Utilizing available custom antisera for AaAQP1, AaAQP4 and AaAQP5, these three aquaporins were localized in each of the organs and the effect of rearing larvae in BW on the expression of these three AaAQPs was assessed. This study provides the first comprehensive report on the potential effects of longterm external salinity on the expression of AaAQPs in osmoregulatory organs of larval mosquitoes. This is of significant importance because *A. aegypti* are increasingly exploiting BW habitats to lay eggs where the larvae then develop into the next generation of disease vectoring adults.

## Materials and methods

### Animal rearing

*Aedes aegypti* eggs (Liverpool) were obtained from a colony reared in the Department of Biology, York University, Toronto, Canada. Eggs were hatched in 800 mL of FW or BW with an additional 6 mL of food solution composed of 1.8 g liver powder and 1.8 g inactive yeast in 500 mL of reverse osmosis water. FW was composed of dechlorinated tap water (Ontario Ministry of Environment: Ontario Drinking Water) and BW was a 30% artificial sea water solution composed of 10.5 g $L^{-1}$ Instant Ocean Sea Salt dissolved in 18.2 MΩm.cm water (approximately equivalent to 144 mmol $L^{-1}$ NaCl). Larvae were kept at room temperature on a 12h:12h light:dark cycle. Larvae were fed daily with 3 mL of food solution. Rearing water was refreshed every two days until larvae reached the 4th instar, upon which point larvae were dissected. The rearing protocol was repeated as described from eggs through to larvae for each biological replicate that was collected.

### Total RNA extraction and cDNA synthesis

Larvae were dissected under physiological saline [5, 31]. 75–80 larvae from a single biological replicate (defined as n = 1) were dissected to isolate GC, MTs, and HG, and each organ type was kept separate and placed in 350 μL of lysis buffer from the PureLink™ RNA Mini Kit (Invitrogen, Carlsbad, California, United States) and β-mercaptoethanol (20%). RNA was extracted according to the PureLink™ RNA Mini Kit instructions. RNA was treated with the TURBO DNAfree™ Kit (Applied Biosystems, Streetsville, Ontario, Canada) to remove genomic DNA following the manufacturer's instructions. The quality and quantity of RNA was determined

**Table 1. Primers and their characterization utilized in qPCR.** Primers for AaAQP1-6 and 18S rRNA were designed and reported previously [15, 19, 33].

| Target | Accession # | Forward Primer | Reverse Primer | Primer Efficiency (%) | Amplicon Size (bp) | Annealing Temperature (°C) |
|--------|-------------|----------------|----------------|----------------------|--------------------|----------------------------|
| AaAQP1 | AF218314.1 | ATCGGATTCAGCAGGAGAG | TGATGTGGCAACCACTTAC | 102.2 | 279 | 58 |
| AaAQP2 | XM_001656882 | TGGCAAAGTCAGCATTGTTC | ACCCAAAGTAACCGTCATGC | 103.8 | 280 | 58 |
| AaAQP3 | XM_001649697 | ATGTTCCGATGCTCATCCTC | TTCTCCCATTTTGCTGTTCC | 100.8 | 251 | 58 |
| AaAQP4 | XM_001650118 | AAGCAACCAGTCGTTTCTG | GAGCATCGGAACATTCAAC | 98.1 | 277 | 58 |
| AaAQP5 | XM_001650119 | GGTGTCCTCGTTCTGGTGTG | CCAACCCAGTAGACCCAGTG | 91.3 | 206 | 58 |
| AaAQP6 | XM_001647996 | ATGCCACTGCTTGTCCCTAC | TTTCCGAAATGACCTTGGAG | 104.7 | 276 | 58 |
| 18S | U65375 | TTGATTCTTGCCGGTACGTG | TATGCAGTTGGGTAGCACCA | 102.1 | 194 | 58 |

using a NanoDrop 2000 spectrophotometer (Thermo Fisher Scientific, Waltham, USA). The quality of RNA was determined based on the ratio between 260:280 nm wavelengths of 2.0 ± 0.2 with the ideal value of 2.0 for RNA. The purified RNA was utilized in the synthesis of cDNA using the iScript™ cDNA Synthesis Kit (Bio-Rad, Mississauga, Ontario, Canada) according to the manufacturer's instructions. The cDNA was stored at -20˚C until subsequent use.

## Quantitative real-time PCR (qRT-PCR)

The primers utilized in the qPCR reactions were designed and described by Marusalin et al., [15] and listed in Table 1. Each reaction consisted of 10µl of SsoFast™ Evagreen® Supermix (Bio-Rad), 2µl of cDNA template, 0.5µL of 10 µmoll⁻¹ forward primer, 0.5µL of 10 µmoll⁻¹ reverse primer and 7µl of DNAse free, RNAse free, sterile PCR water (Invitrogen). Reactions were performed in two technical replicates with a no cDNA template control included for each gene in the CFX96™ real-time PCR detection system (Bio-Rad). Reactions ran with the following settings, 2 min of enzyme activation at 95˚C, followed by 39 cycles of 5s denaturation at 95˚C and 5s of annealing/extension at 58˚C. In order to confirm the presence of a single product, a melting curve analysis was performed after each reaction and consisted of holding the product at 95˚C for 10s followed by holding the product at 65–95˚C in 0.5˚C increments, held for 5s each. For each gene and organ assessed, a standard curve was generated to optimize reaction efficiency. Quantification of transcripts was determined using the Pfaffl method [32]. The 18S rRNA gene was used as the reference gene due to its consistent levels of transcript expression under FW and BW treatments [33]. Primers for 18S rRNA were designed and described by Jonusaite et al. [33] and are included in Table 1.

## Immunohistochemistry

Immunohistochemistry of the GC, HG, and MTs localizing AaAQP1, AaAQP4, AaAQP5, and V-type H⁺ ATPase (VA) was conducted following a previously published and detailed protocol [19]. Briefly, FW and BW-reared *A. aegypti* larvae were obtained for fixation in Bouin's fixative. Tissues were dehydrated and embedded into paraffin wax. Samples were sectioned on a microtome (Leica Microscopy Inc. RM 2125RT manual rotary, ON, Canada) and placed onto slides for subsequent rehydration and antibody probing. Sections of each organ from FW and BW larvae were placed side by side on the same slide such that they were then treated identically throughout the immunohistochemical protocol. Anti-AaAQP1 affinity purified rabbit polyclonal antibody (1.3 µgmL⁻¹ antibody, antigen sequence, CFFKVRKGDEESYDF, Genscript, NJ, USA), anti-AaAQP4 affinity purified rabbit polyclonal antibody (1.0 µgmL⁻¹ antibody, antigen sequence, PAEQAPSDVGKSNQS, Genscript, NJ, USA), or anti-AaAQP5 affinity purified rabbit polyclonal antibody (2.6 µgmL⁻¹ antibody, antigen sequence,

FRREVPEPEYNRELT, Genscript, NJ, USA), in combination with membrane marker guinea pig polyclonal anti-VA (kind donation from Dr. Weiczorek, Germany, 1:15000 dilution in antibody dilution buffer (10% goat serum, 3% bovine serum albumin, 0.05% Triton X100 in phosphate buffered saline) were used to probe the rehydrated tissue samples. A goat anti-rabbit antibody conjugated to AlexaFluor 594 (Jackson Immunoresearch) was applied at 1:500 dilution to visualize AaAQP1, AaAQP4, and AaAQP5, while goat anti-guinea pig antibody conjugated to AlexaFluor 488 (Jackson Immunoresearch) was applied at 1:500 dilution to visualize VA. Negative control slides were also processed as described above with all primary antibodies omitted. Slides were mounted using ProLong® Gold antifade reagent with DAPI (Life Technologies, Burlington, ON, Canada). Fluorescent images were captured using an Olympus IX81 inverted fluorescent microscope (Olympus Canada, Richmond Hill, ON, Canada) and CellSense® 1.12 Digital Imaging software (Olympus Canada).

## Protein processing, electrophoresis, and western blotting

GC, HG, and MTs were isolated under physiological saline from 75–80 *A. aegypti* larvae from each biological replicate reared in FW or BW (defined as n = 1) and stored at—80˚C until processing. Samples were sonicated 3 X 10s at 3.5W using an XL 2000 Ultrasonic Processor (Qsonica) in TRIS homogenization buffer (50 mmolL$^{-1}$ Tris-HCl, pH 7.4 and 1:200 protease inhibitor cocktail; Sigma-Aldrich). Homogenates were then centrifuged at 13,000 g for 15 min at 4˚C [Sorvall™ Legend™ Micro 21 Centrifuge (ThermoFisher Scientific, USA)]. The protein concentration of the supernatant was measured using the DC protein assay (Bio-Rad) according to the manufacturer's guidelines with bovine serum albumin (BSA) as standards. Measurements were carried out on a Multiskan spectrum spectrophotometer (Thermo Electro Corporation, USA) at 750 nm.

Samples were prepared for sodium dodecyl sulphate polyacrylamide gel electrophoresis (SDS-PAGE) by heating for 5 min at 100˚C with 6X loading buffer [225 mmolL$^{-1}$ Tris-HCl, pH 6.8, 3.5% (w/v) SDS, 35% glycerol, 12.5% (v/v) β-mercaptoethanol and 0.01% (w/v) Bromophenol Blue]. 10 µg of protein were loaded onto a 4% stacking and 12% resolving SDS-PAGE gel. Electrophoresis was carried out at 110 V for 1.5 h. Proteins were transferred onto a polyvinyl difluoride (PVDF) membrane using a wet transfer method. Proteins were transferred from the gel at 100 V for 1 h in a cold (∼4˚C) transfer buffer (0.225 g Tris, 1.05 g glycine in 20% methanol). The PVDF membrane was then blocked with 5% skimmed milk powder in TBS-T for 1 h at RT and incubated overnight at 4˚C with the appropriate antibody (Table 2). The PVDF membrane was washed in Tris-buffered saline (TBS-T; 9.9 mmolL$^{-1}$ Tris, 0.15 mmolL$^{-1}$ NaCl, 0.1 mmolL$^{-1}$ Tween-20, 0.1 mmolL$^{-1}$ NP-40, pH 7.4) for 15 min and incubated with horseradish peroxidase (HRP)-conjugated goat anti-rabbit antibody (1:5000 in 5% skim milk in TBS-T) (Bio-rad) for 1 h at RT. PVDF membrane was washed three times for 15 min with TBS-T before carrying out a chemiluminescent reaction using the Clarity™ Western ECL substrate (Bio-Rad). After visualization of detected protein bands, the PVDF membrane

**Table 2. Antibody concentration used for western blotting.** Antibodies were diluted in 5% skimmed milk and TBS-T.

|  | AaAQP1 (µgmL$^{-1}$) | AaAQP4 (µgmL$^{-1}$) | AaAQP5 (µgmL$^{-1}$) |
|---|---|---|---|
| **GC** | 0.32 | 1.04 | 0.64 |
| **HG** | 0.16 | 1.04 | 2.57 |
| **MT** | 0.64 | 0.70 | 1.28 |

GC: gastric caeca, HG: hindgut, MT: Malpighian tubules.

was washed with TBS-T for 1 min before probing the membranes for total protein with Coomassie blue as a loading control as described by Welinder and Ekblad [34]. The Image J 1.49 software (USA) was used to quantify the protein abundance of the western blots. All treatment group densitometry ratios were normalized to the total protein density.

### Statistical analysis

Data were analyzed using Prism® 5.03 (GraphPad Software Inc., La Jolla, California, USA) and expressed as mean values ± standard error of the mean (SEM). The normalized data for qPCR and western blot were analyzed using unpaired t-test or Mann Whitney test to assess the effects of BW treatment in relation to FW for each AaAQP in each organ.

### Results

#### Quantitative expression of AaAQP1, AaAQP4 and AaAQP5 in GC, MTs and HG of FW and BW reared larvae

Larvae that developed in FW or BW had similar levels of AaAQP transcripts in the GC, MTs and HG (Fig 1). Protein homogenates of organs run on western blots and probed with an antibody raised against a specific sequence in the AaAQP1 protein displayed protein bands with different masses (Fig 2 and S1 Fig). The predicted mass of the AaAQP1 monomer is 26.2 kDa. In the GC a band of ~50 kDa was prominent representing a putative dimer. In the MT and HG a band of ~23 kDa was detected representing the putative monomer. The abundance of AaAQP1 was higher in the GC of BW reared larvae relative to FW reared larvae, whereas there was no difference in AaAQP1 protein abundance in the MT and HG between BW and FW reared larvae.

There are three predicted splice variants of AaAQP4 with masses of 31.3, 29, and 26.9 kDa. The protein homogenates of the three organs examined in this study showed bands at ~57 and 58 kDa when probed with an antibody targeting AaAQP4 corresponding to a putative dimer (Fig 3 and S2 Fig). In MTs and HG samples there was also bands observed at ~24 to 26 kDa range which correspond to putative monomers. GC samples revealed bands in the range of ~30 to 35 kDa which correspond to putative monomers with putative post translational modifications (Fig 3A). HG samples revealed bands between 25 and 50 kDa but the 57–58 kDa band consistently appeared throughout all samples (Fig 3C). There was no difference in the protein abundance of AaAQP4 of FW and BW reared larvae in the organs examined.

The predicted mass of the AaAQP5 protein monomer is 26.4 kDa. The protein homogenates of the organs studied showed bands of ~ 23–32 kDa when probed with AaAQP5 antibody representing putative monomers (Fig 4 and S3 Fig). A band of ~55 kDa was also detected in homogenates of GC representing a putative dimer, and some homogenates of MTs revealed bands of 60 and 75 kDa representing putative oligomers. Rearing larvae in BW did not affect AaAQP5 protein abundance in any of the organs assessed relative to those of larvae reared in FW.

#### Immunolocalization of AaAQP1, AaAQP4 and AaAQP5 in osmoregulatory organs of *Aedes aegypti* larvae and the effects of salinity

A thorough analysis of the expression and localisation of the V-type H⁺-ATPase (VA) in osmoregulatory organs of *A. aegypti* larvae has been previously reported [35]. We utilized VA immunoreactivity as a marker to aid in determining the localization of AaAQP1, AaAQP4 and AaAQP5 in the larvae of *A. aegypti*.

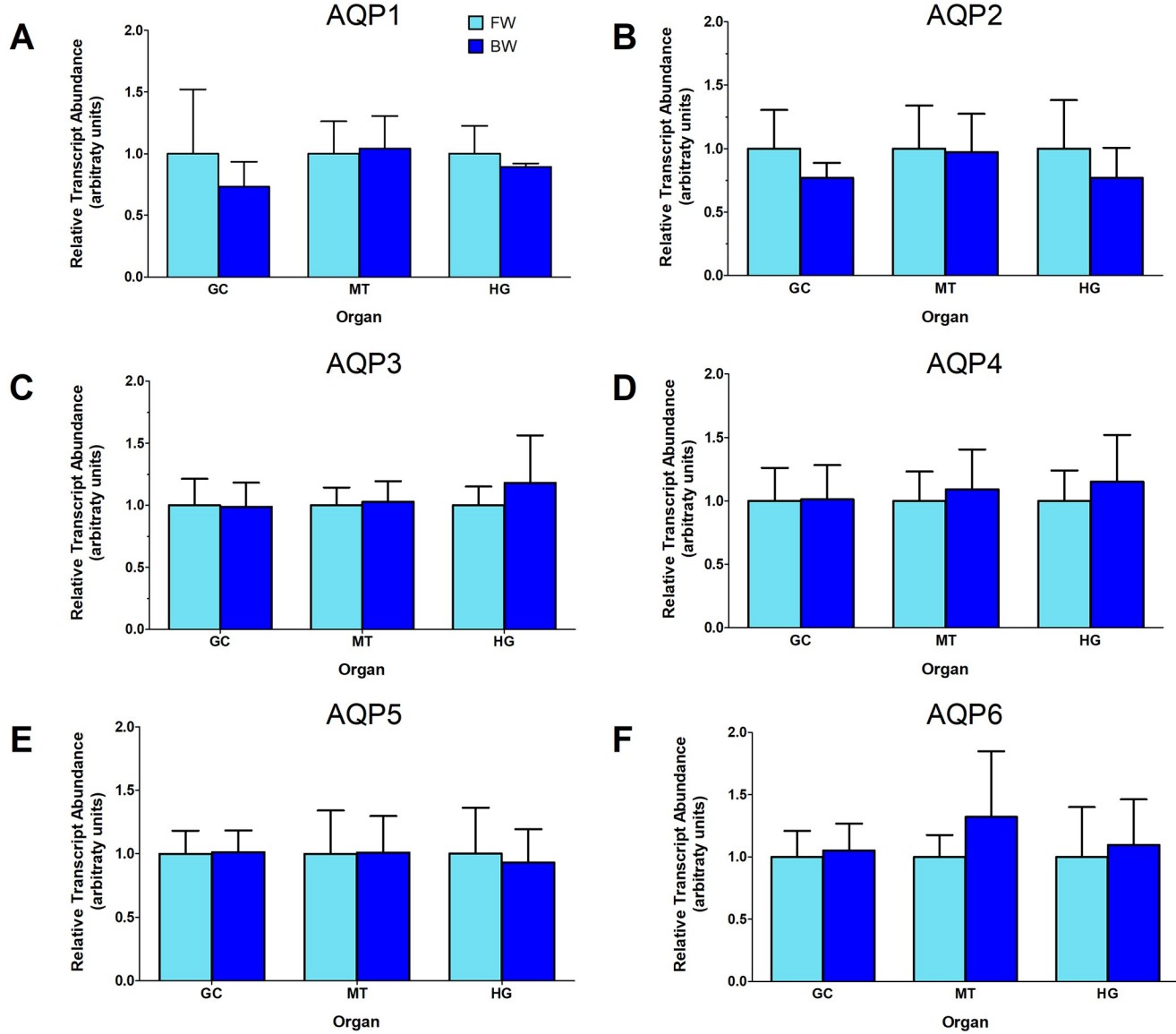

**Fig 1. mRNA abundance of aquaporin (AQP) genes in the osmoregulatory organs of larval *Aedes aegypti* of freshwater (FW) reared larvae (dechlorinated tapwater, light blue bars) and brackish water (BW) reared larvae (30% seawater, dark blue bars).** AaAQP1 (A), AaAQP2 (B), AaAQP3 (C), AaAQP4 (D), AaAQP5 (E), and AaAQP6 (F) transcript abundance was measured using quantitative-PCR, normalized to 18S ribosomal RNA and expressed as fold change relative to freshwater. Data are expressed as means ± SEM and statistically analyzed using unpaired t-tests with differences $p \leq 0.05$. GC: gastric caeca, MT: Malpighian tubules, HG: hindgut; n = 5 except for GC-FW and MT-FW where n = 4.

**Gastric caeca.** AaAQP1, AaAQP4, AaAQP5, and V-type H⁺-ATPase (VA) immunoreactivity was detected in the GC of larval *A. aegypti* (Fig 5). In FW reared larvae, the VA was localized to the distal portion of the GC containing ion transporting cells (Fig 5A, solid arrow). The digestive cells at the proximal region of the GC did not exhibit VA immunoreactivity above background (Fig 5A,5C and 5E). In BW, VA was also localized to the apical (lumen facing) membrane of the cells which are no longer regionally segregated under BW conditions (Fig 5D, solid arrow). In FW reared larvae, AaAQP1 immunoreactivity localized to the apical membrane of digestive cells in the proximal region of the GC (Fig 5A, dashed arrow). In BW,

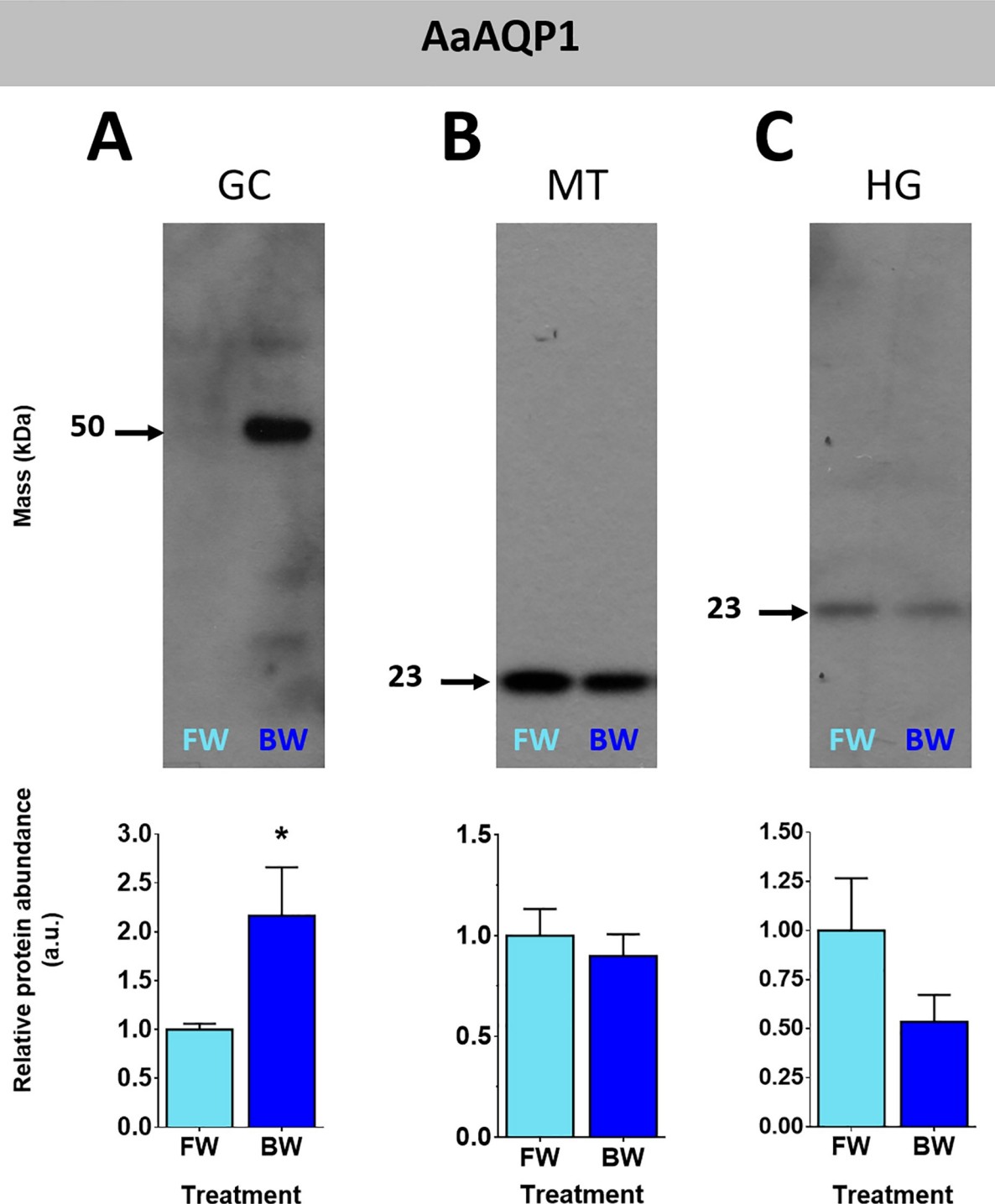

**Fig 2. Protein abundance of AaAQP1 in organs of *A. aegypti* larvae reared in freshwater or brackish water.** The gastric caeca (A), Malpighian tubules (B), and hindgut (C), of freshwater (FW, light blue bars) and brackish water (BW, 30% seawater, dark blue bars) reared *Aedes aegypti* larvae. Individual band densities were normalized to total protein assessed using Coomassie staining of total protein. The brackish water densitometry was expressed relative to freshwater. Data are expressed as means ± SEM and statistically analyzed using unpaired t-tests where a significant difference was p ≤ 0.05 indicated by *, n = 5. GC: gastric caeca, MT: Malpighian tubules, HG: hindgut.

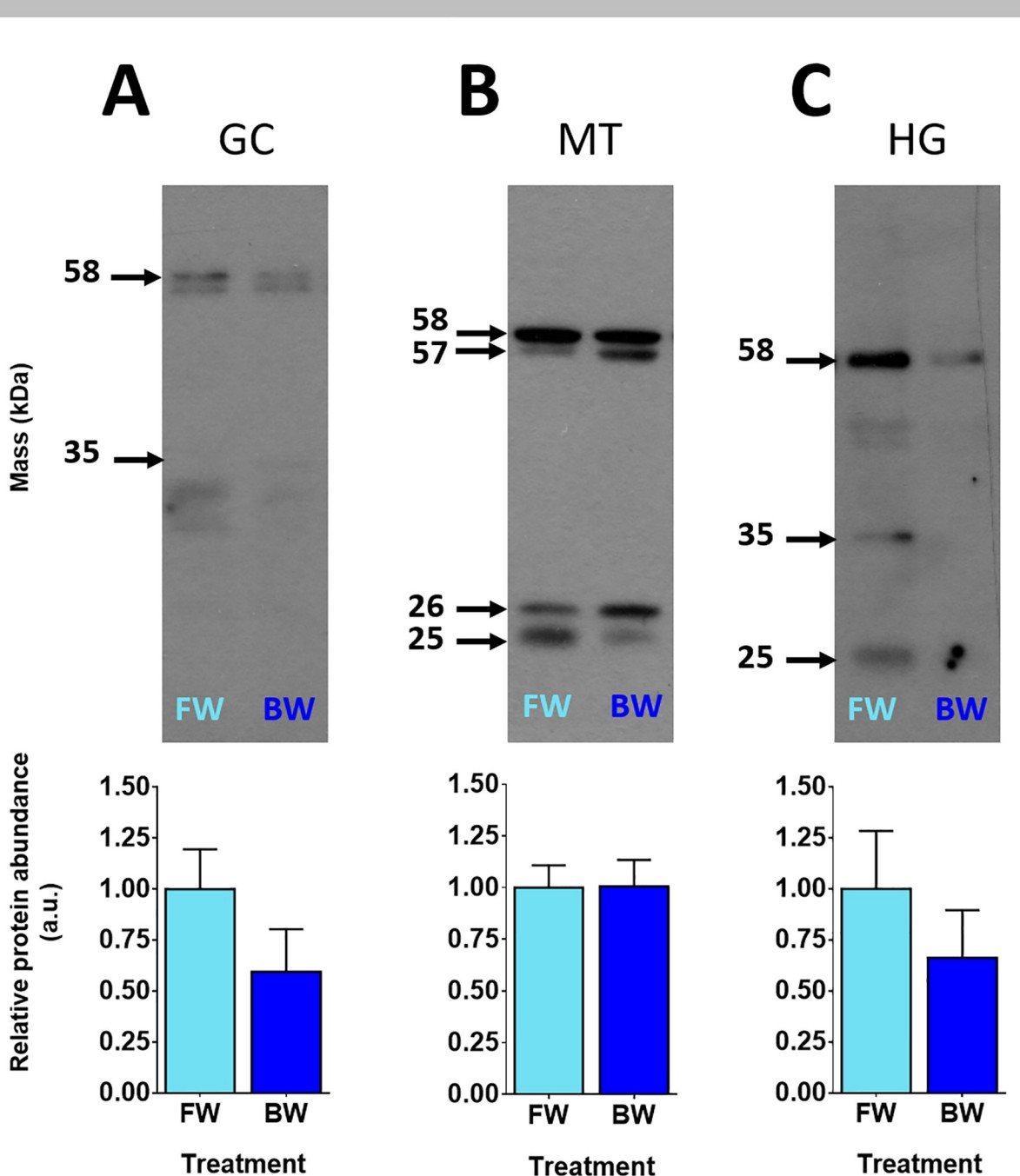

**Fig 3. Protein abundance of AaAQP4 in organs of mosquito larvae.** The gastric caeca (A), Malpighian tubules (B) and hindgut (C) of freshwater (FW, light blue bars) and brackish water (BW, 30% seawater, dark blue bars) reared larval *Aedes aegypti*. Individual band densities were normalized to total protein assessed using Coomassie staining. Brackish water densitometry was expressed relative to freshwater. Data are expressed as means ± SEM and statistically analyzed using unpaired t-tests where a significant difference was p ≤ 0.05, n = 5. GC: gastric caeca, MT: Malpighian tubules, HG: hindgut.

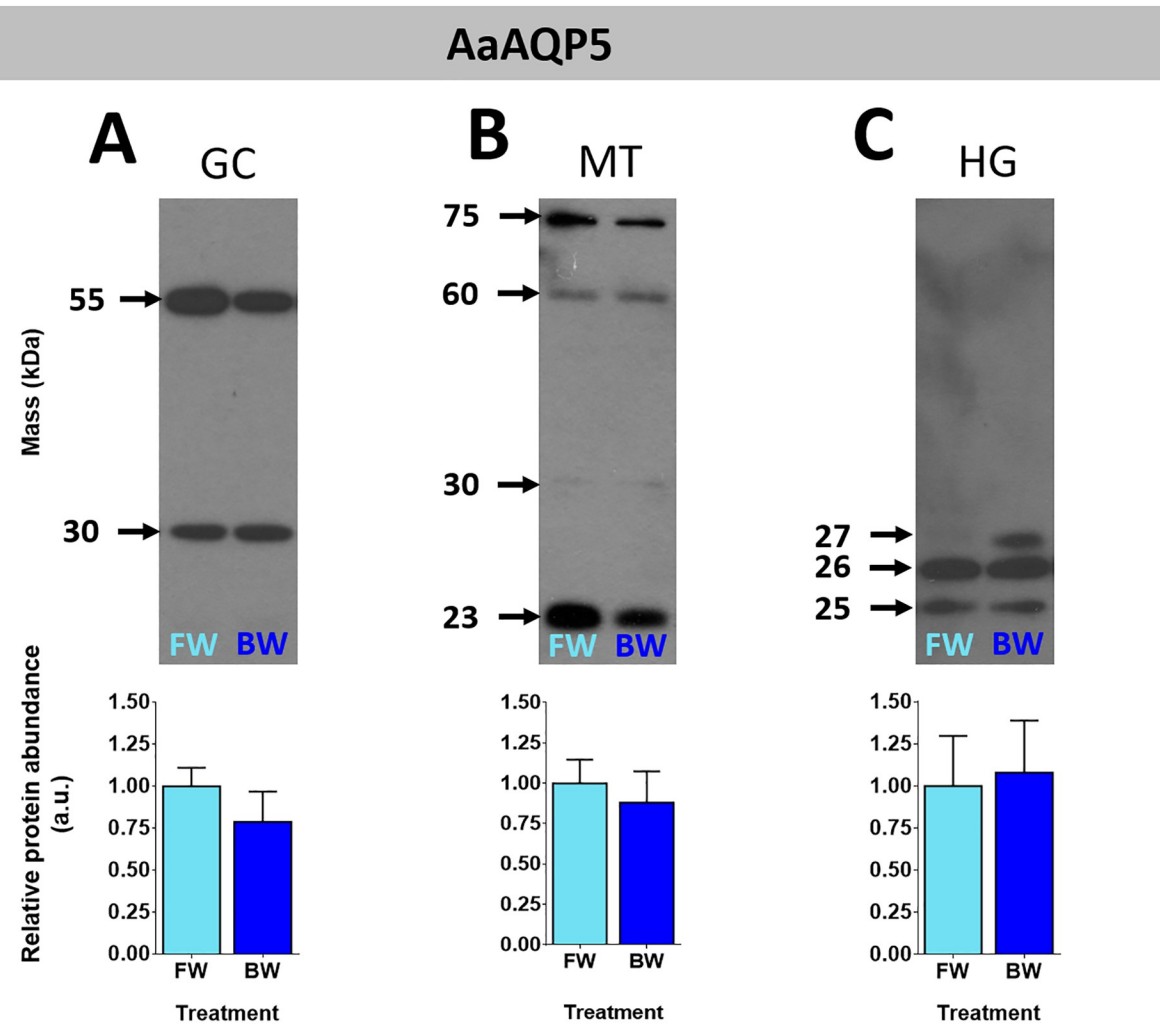

**Fig 4. Protein abundance of AaAQP5 in organs of *Aedes aegypti* larvae reared in freshwater or brackish water.** The gastric caeca (A), Malpighian tubules (B) and hindgut (C) of freshwater (FW, light blue bars) and brackish water (BW, 30% seawater, dark blue bars) reared *Aedes aegypti* larvae. Individual band densities were normalized to total protein assessed using Coomassie staining. Brackish water densitometry was expressed relative to freshwater. Data are expressed as means ± SEM and statistically analyzed using unpaired t-tests where a significant difference was p ≤ 0.05, n = 5. GC: gastric caeca, MT: Malpighian tubules, HG: hindgut.

AaAQP1 was also apparent in the apical membrane regions of cells throughout the GC where it co-localized with VA staining (Fig 5B, dashed arrows).

Immunolocalization of AaAQP4 in the GC of FW reared larval *A. aegypti* revealed diffuse immunoreactivity throughout the ion transporting cells which co-localized with VA immunoreactivity (Fig 5C, yellow stain). AaAQP4 immunoreactivity in the digestive cells of the GC without specific membrane localization was also observed (Fig 5C, red stain, dashed arrow). A qualitative reduction in AaAQP4 immunoreactivity was observed in the GC of BW reared larvae (Fig 5D). AaAQP4 immunoreactivity was present in the fat body cells of both FW and BW reared larvae (Fig 5C, arrowhead).

Immunolocalization of AaAQP5 in the GC of larvae was localized to the basolateral membrane of ion transporting and digestive cells in FW and BW reared larvae (Fig 5E and 5F, dashed arrows). In FW larvae, the AaAQP5 staining was co-localized with VA staining on the

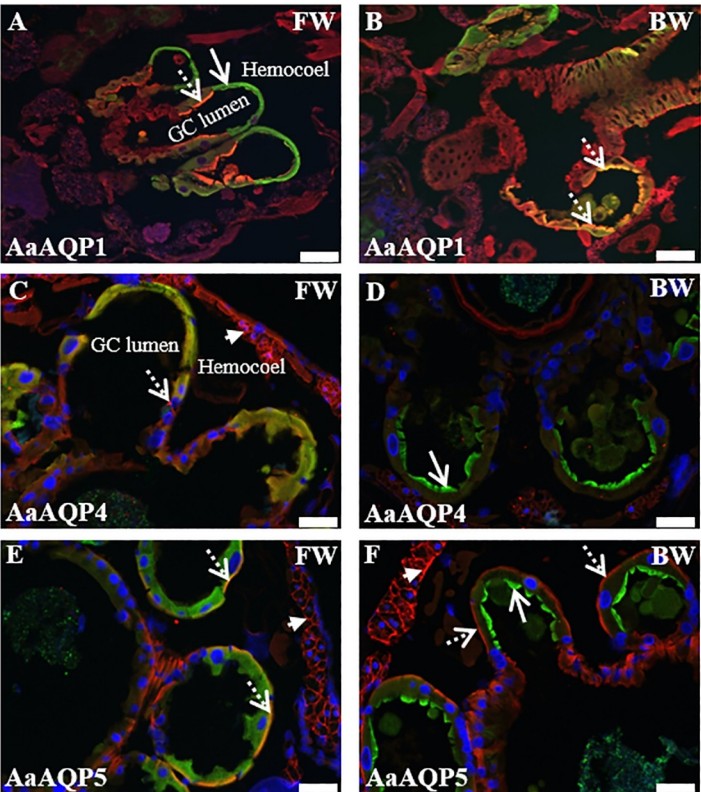

**Fig 5. Immunolocalization of AaAQP1, AaAQP4 and AaAQP5 (red stain) in the gastric caeca (GC) of larval *A. aegypti* reared in freshwater and brackish water using V-type H⁺-ATPase (green stain) as a marker.** Representative paraffin-embedded sections through the frontal plane (A,B) and the transverse plane (C,D,E,F) of *A. aegypti* larvae showing GC. Dashed arrow in A indicates AaAQP1 apical staining. Solid arrow in A indicates VA staining. Dashed arrows in B indicate apical co-localization of AaAQP1 and VA staining (yellow). Dashed arrows in C indicate AaAQP4 staining. Dashed arrows in E and F point to AaAQP5 basal staining. Solid arrows in D and F indicate apical VA staining. Arrowheads in C and E indicate AaAQP4 and AaAQP5 staining in fat body, respectively. DAPI nuclear staining is shown in blue. Scale bar 100 μm for A -B; 50 μm for C-F.

basolateral membrane of ion transporting cells (Fig 5E, dashed arrows, yellow stain). AaAQP5 immunoreactivity was also apparent in fat body cells of FW and BW reared larvae (Fig 5E and 5F, arrowhead).

**Malpighian tubules and hindgut.** Immunoreactivity to AaAQP1, AaAQP4, AaAQP5 and VA antibodies was detected in the HG and MTs of *A. aegypti* larvae (Fig 6) with the exception that there was no detectable AaAQP1 staining in the HG (Fig 6A and 6B). VA immunoreactivity was primarily localized to the apical (lumen side) of the MTs and HG (green and yellow staining in Fig 6A and 6B). In the MTs the VA immunoreactivity also stained the cytosolic region of the cells. Immunoreactivity to AaAQP1 was co-localized with VA to the apical side of the MTs principal cells (yellow and orange stain, Fig 6A and 6B). In cross sections of MTs punctate AaAQP1 immunoreactivity was also observed in some places on the basal side of the MT epithelium which may be either stellate cells or tracheolar cells (dashed arrows, Fig 6A). There was no discernable difference in AaAQP1 immunoreactivity in the MTs of FW versus BW reared larvae.

AaAQP4 immunoreactivity in MTs was predominantly co-localized with VA to the apical side of the principal cells (dashed arrows, Fig 6C and 6D) but there was also some staining on the basal side (solid arrows, Fig 6C and 6D). There was no qualitative difference in AaAQP4

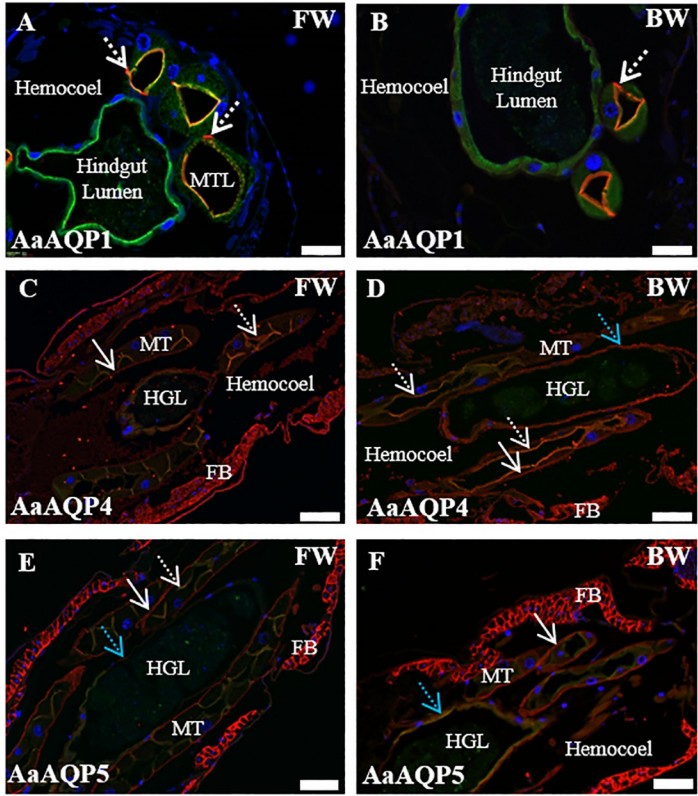

**Fig 6. Immunolocalization of AaAQP1, AaAQP4 and AaAQP5 (red stain) in the Malpighian tubules (MT) and hindgut of larval *A. aegypti* reared in freshwater and brackish water using V-type H+-ATPase (VA, green stain) as a marker.** Representative paraffin-embedded sections through the transverse plane (A,B) and the frontal plane (C,D,E, F) of *A. aegypti* larvae showing the MT and hindgut. Dashed arrows in A and B indicate AaAQP1 staining in apparent stellate cells or trachea. Yellow and orange staining in A and B are areas of co-localization of AaAQP1 and VA. Dashed arrows in C and D indicate co-localization of AaAQP4 and VA staining. Solid arrows in C and D indicate basal AaAQP4 staining. Blue arrow in D indicates AaAQP4 staining in hindgut. Dashed arrow in E indicates apical co-localization of AaAQP5 and VA staining. Solid arrows in E and F indicate basal AaAQP5 staining. Blue arrow in E and F indicate co-localization of AaAQP5 and VA staining in hindgut. MTL = Malpighian tubule lumen; HGL = hindgut lumen; FB = fat body. DAPI nuclear staining is shown in blue. Scale bar 50 μm for A and B; 100 μm for C, D, E and F.

immunoreactivity in the MTs of FW versus BW reared larvae. In BW reared larvae AaAQP4 immunoreactivity was apparent and appeared to be predominantly basal in the HG (blue arrow, Fig 6D). In FW reared larvae the AaAQP4 immunoreactivity in the HG appeared less robust than that present in the BW reared larvae (Fig 6C). AaAQP4 immunoreactivity was evident in the cells of the fat body (FB, Fig 6C).

AaAQP5 immunoreactivity in the MTs of *A. aegypti* larvae was previously reported by Misyura et al. where it was localized to both apical and basal sides of principal cells [24]. In the current study AaAQP5 immunoreactivity was predominantly basal (solid arrow, Fig 6E and 6F) with areas of co-localization with apical VA (dashed arrows, Fig 6E). In the MTs of BW reared larvae basal staining of AaAQP5 was clearly evident; however, apical staining was apparently reduced (Fig 6F). In the HG of FW reared larvae AaAQP5 staining was limited to some regions and appeared faint (blue arrow, Fig 6E) while in BW reared larvae AaAQP5 staining was present and predominantly co-localized with apical VA (blue arrow, Fig 6F).

## Discussion

### Overview

The mosquito *Aedes aegypti* is established in tropical and subtropical regions of the world where it is a vector for Chikungunya, dengue, yellow fever and Zika. Larvae develop in small temporary pools of water like tree holes and human made containers and the larvae have been found in FW as well as BW [2, 3]. Each of these habitats represent unique challenges to the regulation of salt and water levels in body fluids. Since aquaporins are essential regulators of water flux across cell membranes and epithelia, we compared the expression of aquaporins in osmoregulatory organs of larvae that developed in either FW or BW in an effort to understand the underlying physiology that allows larvae to inhabit these different aquatic environments.

**Aquaporin protein expression in osmoregulatory organs.** Western blotting for AaAQPs produced signals with different masses both within an organ and between different organs. Multiple signals in western blot of insect organ homogenates probed with custom antibodies against aquaporins have been reported [19, 24, 36]. In general, there was at least one signal shared among all organ homogenates for each antibody (e.g. AaAQP1, AaAQP4 and AaAQP5). Further discussion on signals resulting from MTs homogenates probed with AaAQP5 is warranted because our own previous studies of larvae that were reared in reverse osmosis water and fed dsRNA showed a putative modified AaAQP5 monomer of ~32 kDa [24]. In this current study a faint ~ 32 kDa signal was observed in MTs homogenates of larvae that developed in FW (dechlorinated tapwater) or BW; however, a signal of ~23 kDa was predominant. The precise reason for multiple and varying signals is not known; however, some plausible explanations are provided. Aquaporins form functional oligomers which can resist dissociation to varying degrees under sodium dodecyl sulfate polyacrylamide gel electrophoresis [37–39]. Aquaporin monomers undergo post-translational modifications like glycosylation and phosphorylation which can alter their mass, and these can form oligomers [29, 40–43]. Two splice variants of at least one mosquito aquaporin, AgAQP1 has been shown to be expressed in an organ specific manner [44]. Therefore, we propose that the variability in the resulting signals of MTs homogenates probed with AaAQP5 antisera results from one or a combination of these reasons.

### Expression of aquaporins in the gastric caeca

The GC epithelium is composed of cells (digestive) that express $Na^+ / K^+$—ATPase (NKA) on the basal membrane and other cells (ion-transporting) that express VA on the basal and apical membranes [20, 22]. When larvae develop in FW the digestive cells are located in the proximal region and the ion transporting cells are in the distal region of the GC, but when larvae develop in BW the majority of the surface area of the GC is populated by NKA-rich digestive cells [20]. Both cell types transport $Na^+$ and $K^+$ from hemolymph to lumen which is likely to drive nutrient absorption [20]. Water flux across the GC epithelium has not been studied; however it has been suggested that in FW, the higher ion-motive enzyme activities (VA and NKA) and ion secretion rates in the GC also serve to support water secretion in the GC since imbibed FW is likely to be absorbed along the surface of the midgut [20]. This would help counter dilution of the hemolymph while preserving adequate levels of luminal fluid to support digestion and nutrient absorption [20]. This theory is plausible for the proximal digestive cells which are shown here to express AaAQP5 and AaAQP1 on the basal and apical membranes, respectively. Here, we propose that the AaAQP1 expressing cells in the GC of larvae that developed in FW are the digestive cells because of their proximal location and the absence of any appreciable VA immunoreactivity. AaAQP1 transports water and AaAQP5 transports water as well as

solutes like trehalose and therefore, proximal cells possess a route for transcellular water flux [30]. On the other hand, distal ion transporting cells express AaAQP5 in the basal membrane and AaAQP4, which is a poor water transporter (when expressed in a heterologous expression system), is primarily cytosolic. Hence, we cannot confirm whether transcellular water flux across ion transporting cells is possible. Protein level studies of AaAQP2 and AaAQP6, both water transporters, are required to clarify this point but antibodies are not available at present. Interestingly, the ortholog of AaAQP1 in larvae of the mosquito *Anopheles gambiae* (AgAQP1) is expressed on the basal side throughout the gastric caeca epithelium where it may mediate water movement between the hemolymph and gastric caeca cells [44].

The basal membrane expression of AaAQP5 by both cell types could facilitate trehalose uptake, the major blood (hemolymph) sugar in insects, required to fuel the ion transporting and digestive functions of the gastric caeca. To this end, the expression of AaAQP4 may also serve for trehalose uptake, although expression appeared to be primarily cytosolic so it is unclear if AaAQP4 could be recruited to the membrane when needed. AaAQP4 and AaAQP5 transcript and protein abundance was the same whether larvae developed in FW or BW suggesting two possibilities. One, the primary function of AaAQP4 and AaAQP5 in the GC is not in osmoregulation or two, these two aquaporins are primarily regulated with post translational modifications rather than the abundance of their expression. Despite there being no difference in transcript abundance, AaAQP1 protein abundance was higher in GC of larvae that developed in BW where it remained on the apical membrane of cells. Under these conditions, imbibed water (30% seawater) is iso-osmotic to the hemolymph and contains significantly more salt which is reflected in the luminal ion composition [20]. In this regard, dilution of the hemolymph is no longer an issue and both ion-motive enzyme activities and ion secretion are lower [20]. The NKA rich digestive cells are now found throughout all regions of the GC and make up the majority of the basal surface area [20]. Coincidentally, AaAQP1 apical expression was now seen throughout the GC further supporting the idea that it is the NKA rich digestive cells that express AaAQP1. The increase in AaAQP1 protein abundance may simply reflect more digestive cells present in the GC of larvae that develop in BW. Under these iso-osmotic conditions AaAQP1 could provide a route for transcellular water flux across the entire surface of the GC allowing for quick compensatory water fluxes and/or sustaining luminal fluid levels for digestion and absorption.

## Expression of aquaporins in Malpighian tubules

The Malpighian tubules have the greatest abundance of AaAQP transcripts which is a reflection of their important secretory function which serves to rid larvae of excess water when they develop in freshwater [10, 15]. AaAQP5 expression was previously mapped to both apical and basal sides of the MTs principal cells, and dsRNA-mediated knockdown of AaAQP5 decreased fluid secretion rates of MTs [24]. Here we show that both entomoglyceroporins, AaAQP4 and AaAQP5, are expressed on the apical and basal membranes of principal cells. In the terrestrial adult *Drosophila*, two entomoglyceroporins are expressed by principal cells and knockdown of either one decreased fluid secretion by MTs [36]. AaAQP1 was the first AQP to be cloned and studied in *Aedes aegypti* and it was shown to be expressed by tracheolar cells in the Malpighian tubules of adult females [29, 45]. Adult female mosquitoes typically need to conserve water and the MTs are relatively inactive prior to blood-feeding; however, when rapid diuresis commences following a blood meal, AaAQP1 is important in clearing water from the tracheal system associated with the MTs such that the sudden high demand for oxygen by MTs epithelial cells can be met [45]. In the adult mosquito *Anopheles gambiae* and adult *Drosophila*, orthologs of AaAQP1 are expressed in the stellate cells of the MTs [36, 46]. Although adult MTs principal

cells do not express AaAQP1, the principal cells of MTs of larvae express AaAQP1 where it localizes to the apical membrane. Furthermore, distinct, punctate—like spots with intense AaAQP1-like immunoreactivity can be seen on the basal side of the MTs cross sections which we suggest could be either stellate cells or tracheolar cells. Unlike adults, larvae need to continuously clear excess water when they are in freshwater and the MTs play a critical role in this process. The differences in expression of AaAQP1 between larvae and adults is therefore not surprising. Substantial differences in the transcriptome of MTs of mosquito larvae and adults have been reported and suggested to be related to the different functions the MTs need to play in larvae versus adults [47].

Immunolocalization and protein abundance of AaAQP1, AaAQP4 and AaAQP5 in the MTs is the same whether larvae develop in FW or BW. This is consistent with the similar secretion rates of isolated, unstimulated MTs from FW and BW reared larvae [5]. Collectively, this suggests that despite the fact that the osmotic gradient for passive water accumulation in the body fluids of the larvae is greatly reduced in BW, and theoretically larvae would not need to eliminate as much water, the MTs maintain their capacity for water flux. Instead, MTs alter the relative transport rates of cations, specifically reducing $K^+$ transport when larvae are in BW, which may then favour removal of $Na^+$ that has accumulated in the hemolymph [5].

The importance of AaAQP5 in water transport by Malpighian tubules of larvae has been shown [24]. AaAQP4 is similarly situated on apical and basal membranes of principal cells where it could also contribute to water flux; however, AaAQP4 was shown to be a poor transporter of water in a heterologous expression system [30]. An alternative, or additional, function of AaAQP4 and AaAQP5 on the basal membrane is for trehalose uptake from the hemolymph to support metabolic functions of MTs cells as suggested for the GC (see above). In addition, AaAQP2 transcript levels in MTs of larvae are substantially high and warrant a thorough investigation at the protein level when an antibody is available [15, 47].

## Expression of aquaporins in hindgut

Aquaporins are expressed in the HG of terrestrial insects where water reabsorption prior to excretion is vital to avoid dehydration [48, 49]. In *A. aegypti* larvae developing in FW, the HG is thought to be responsible for reabsorption of ions and not water, thereby resulting in a dilute urine which serves to conserve ions while eliminating excess water. It is interesting that transcripts of AaAQPs are present in the HG but admittedly, at low levels [15]. We detected putative AaAQP1, AaAQP4 and AaAQP5 protein signals in western blots. Despite the detection of signal in western blots of the HG there was no detectable AaAQP1-like immunoreactivity in either FW or BW larvae. Since HG protein homogenates were pooled from at least 75 larvae it is possible that enough protein was isolated for detection in western blots; however, immunohistochemistry on individual HG sections suggest there is very little expression of AaAQP1 rendering the signal undetectable. AaAQP4 and AaAQP5 protein abundance was similar in HG of FW and BW larvae; however, both AaAQP4 and AaAQP5-like immunoreactivity appeared more intense in BW with AaAQP4 and AaAQP5 on the basal and apical sides of the epithelium, respectively. If AaAQP4 conducts water then there is a pathway for water flux across the HG epithelium; however, these AQPs also conduct solutes such as trehalose which may be their principal function. Therefore, our results are largely consistent with the idea that the HG of *A. aegypti* larvae is relatively impermeable to water.

## Conclusions

Aquaporin expression was examined in the GC, MTs and HG of *A. aegypti* larvae that developed in hypo-osmotic FW and iso-osmotic BW. The water transporting AaAQP1 was shown

to be apically expressed while AaAQP4 and AaAQP5, which are better conductors of solutes like trehalose than water, can be expressed both apically and basally. Overall, few alterations in AaAQP expression were seen in these organs when comparing larvae that developed in FW or BW suggesting that AaAQPs are functionally important under both hypo- and iso-osmotic conditions. AaAQPs may also be preferentially regulated through post-translational modifications evidenced by the detection of putative modified monomers and oligomers in western blots. The greatest potential for functional changes are seen in the GC and the anal papillae [19], both of which are in contact with the external medium, the GC from water imbibed by the larvae and the anal papillae as externally projecting organs. Investigation of AaAQP expression in other organs like the midgut and fatbody warrant attention, as well as protein level studies on AaAQP2 and AaAQP6 when specific antibodies become available. Furthermore, studies utilizing dsRNA to knockdown expression of select AaAQPs, such as those already performed for AaAQP5 in the MTs, will shed more light on AaAQP function. Lastly, similar studies examining short term, abrupt effects of salt water on AaAQP expression and function may provide further insight and would mimic what may occur in the mosquito larvae's natural habitat if abrupt salinization of freshwater occurs.

## Supporting information

**S1 Fig. Exposed film of western blots of gastric caeca (A), Malpighian tubules (B) and hindgut (C) protein homogenates probed with AaAQP1 antibody (top row of images) and their respective blots with total protein illustrated with coomassie blue staining (bottom row of images).** Five biological replicates of organ protein homogenates from freshwater reared larvae are loaded in lanes 1, 3, 5, 7 and 9. Five biological replicates of organ protein homogenates from brackish water reared larvae are loaded in lanes 2, 4, 6, 8 and 10. The red rectangle indicates the lanes shown in Fig 2. For protein quantification of AaAQP1 the ~ 23 kDa band was used for MT and HG and the ~50 kDa band was used for GC. L = Ladder; GC = gastric caeca; MT = Malpighian tubules; HG = Hindgut.
(TIF)

**S2 Fig. Exposed film of western blots of gastric caeca (A), Malpighian tubules (B) and hindgut (C) protein homogenates probed with AaAQP4 antibody (top row of images) and their respective blots with total protein illustrated with coomassie blue staining (bottom row of images).** Five biological replicates of organ protein homogenates from freshwater reared larvae are loaded in lanes 1, 3, 5, 7 and 9. Five biological replicates of organ protein homogenates from brackish water reared larvae are loaded in lanes 2, 4, 6, 8 and 10. The red rectangle indicates the lanes shown in Fig 3. For quantification of AaAQP4 protein the band intensities of the putative monomers and dimers were summed. L = Ladder; GC = gastric caeca; MT = Malpighian tubules; HG = Hindgut.
(TIF)

**S3 Fig. Exposed film of western blots of gastric caeca (A), Malpighian tubules (B) and hindgut (C) protein homogenates probed with AaAQP5 antibody (top row of images) and their respective blots with total protein illustrated with coomassie blue staining (bottom row of images).** Five biological replicates of organ protein homogenates from freshwater reared larvae are loaded in lanes 1, 3, 5, 7 and 9. Five biological replicates of organ protein homogenates from brackish water reared larvae are loaded in lanes 2, 4, 6, 8 and 10. The red rectangle indicates the lanes shown in Fig 3. Putative monomers and dimers were used for protein quantification. L = Ladder; GC = gastric caeca; MT = Malpighian tubules; HG = Hindgut.
(TIF)

**S1 Data.**
(XLSX)

**S2 Data.**
(XLSX)

**S3 Data.**
(XLSX)

## Acknowledgments

The authors thank Dr Jean-Paul Paluzzi and Dr. Scott P. Kelly for advice and suggestions throughout the completion of this study and Amanda Jass for helping with some of the organ dissections.

## Author Contributions

**Conceptualization:** Lidiya Misyura, Andrew Donini.

**Data curation:** Lidiya Misyura, Andrew Donini.

**Formal analysis:** Lidiya Misyura, Andrew Donini.

**Funding acquisition:** Andrew Donini.

**Investigation:** Lidiya Misyura, Elia Grieco Guardian, Andrea Claire Durant.

**Methodology:** Lidiya Misyura, Andrew Donini.

**Project administration:** Andrew Donini.

**Resources:** Andrew Donini.

**Supervision:** Andrew Donini.

**Writing – original draft:** Lidiya Misyura.

**Writing – review & editing:** Elia Grieco Guardian, Andrea Claire Durant, Andrew Donini.

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
