## [Decision Letter · Decision Letter 0]

24 Jun 2020

PONE-D-20-16879

A comparison of aquaporin expression in mosquito larvae (Aedes aegypti) that develop in hypo-osmotic freshwater and iso-osmotic brackish water.

PLOS ONE

Dear Dr. Donini,

Thank you for submitting your manuscript to PLOS ONE. After careful consideration, we feel that it has merit but does not fully meet PLOS ONE’s publication criteria as it currently stands. Therefore, we invite you to submit a revised version of the manuscript that addresses the points raised during the review process.

We look forward to receiving your revised manuscript.

Kind regards,

Dmitri Boudko

Academic Editor

PLOS ONE

Journal Requirements:

Reviewers' comments:

Reviewer's Responses to Questions

**Comments to the Author**

1. Is the manuscript technically sound, and do the data support the conclusions?

Reviewer #1: Yes

2. Has the statistical analysis been performed appropriately and rigorously? 

Reviewer #1: Yes

3. Have the authors made all data underlying the findings in their manuscript fully available?

Reviewer #1: Yes

4. Is the manuscript presented in an intelligible fashion and written in standard English?

Reviewer #1: Yes

5. Review Comments to the Author

Reviewer #1: This study examined expression of aquaporins in Aedes larvae reared under brackish and fresh water conditions. The general conclusion is that there is little change in the AQP levels, which suggests that there is little role in AQPs in relation to adapting to brackish waters.

My only major concern is without functional studies, such as RNAi, that that role of AQPs shoudl not be discounted. It might be worth mentioning there could be a role post-translational. The AQPs could have a role and expressional changes in other tissues.

Also, prolonged rearing of the larvae udner the conditions might have missed the changes that occurred earlier in larval development (worth mentioning, but not a major issue).

Otherwise, I have few other comments.

Specific points:

1. Check references

2. Double check figure references in text.

3. I would suggest combining Figure 2-4 into a single figure (since most show no significance).

4. Arrow labeling isn't consistent on Figure 5.

5. qPCR: Please provide evidence that 18s RNA (control).

6. Provide df and F values for stats.

6. PLOS authors have the option to publish the peer review history of their article (what does this mean?). If published, this will include your full peer review and any attached files.

Reviewer #1: No

---

## [Author Response · Author response to Decision Letter 0]

26 Jun 2020

Reviewer #1: This study examined expression of aquaporins in Aedes larvae reared under brackish and fresh water conditions. The general conclusion is that there is little change in the AQP levels, which suggests that there is little role in AQPs in relation to adapting to brackish waters.

My only major concern is without functional studies, such as RNAi, that that role of AQPs shoudl not be discounted. It might be worth mentioning there could be a role post-translational. The AQPs could have a role and expressional changes in other tissues.

Also, prolonged rearing of the larvae udner the conditions might have missed the changes that occurred earlier in larval development (worth mentioning, but not a major issue).

Author Response: These are good points. We have already seen drastic changes in anal papillae in our previous studies. Other areas of the GI tract could also play a role (e.g. midgut) as well as the fatbody which requires further investigation. We have modified the conclusion section at the end of the Discussion to address these points.

Otherwise, I have few other comments.

Specific points:

1. Check references

Author Response: Thank you. We have examined the reference section and did spot several errors which have now been corrected.

2. Double check figure references in text.

Author Response: We have reviewed our reference to figures in the text and could not find any errors.

3. I would suggest combining Figure 2-4 into a single figure (since most show no significance).

Author Response: Good suggestion. We had previously combined all these blots into a very large multi-panel figure but we felt this created confusion in terms of which aqp protein the images and graphs were corresponding to. We therefore elected to separate them and feel that this makes it much clearer for readers.

4. Arrow labeling isn't consistent on Figure 5.

Author Response: We have carefully reviewed the arrow labels on Figure 5 and how these are explained in the figure caption and how they are explained in the results text of the manuscript. We could not find any errors in these labels and how they are explained.

5. qPCR: Please provide evidence that 18s RNA (control).

Author Response: Thank you. The expression values for 18S in all organs of FW and BW reared larvae of this study are provided in the supplementary file for qPCR data. The validity of using 18s RNA was also previously shown by Jonusaite et al 2016. We have now included this reference with the statement about 18s RNA in the Methods section.

6. Provide df and F values for stats.

Author Response: We have now included an excel spreadsheet with all of the statistical values for the qPCR and western blot quantitative data. The spreadsheet is provided with supplementary data.

---

## [Decision Letter · Decision Letter 1]

4 Aug 2020

A comparison of aquaporin expression in mosquito larvae (Aedes aegypti) that develop in hypo-osmotic freshwater and iso-osmotic brackish water.

PONE-D-20-16879R1

Dear Dr. Donini,

We’re pleased to inform you that your manuscript has been judged scientifically suitable for publication and will be formally accepted for publication once it meets all outstanding technical requirements.

Cheers,

Dmitri Boudko. PhD

Academic Editor

PLOS ONE

---

## [Editor Report · Acceptance letter]

7 Aug 2020

PONE-D-20-16879R1 

A comparison of aquaporin expression in mosquito larvae (Aedes aegypti) that develop in hypo-osmotic freshwater and iso-osmotic brackish water. 

Dear Dr. Donini:

I'm pleased to inform you that your manuscript has been deemed suitable for publication in PLOS ONE. Congratulations! Your manuscript is now with our production department. 

Kind regards, 

on behalf of

Dr. Dmitri Boudko 

Academic Editor

PLOS ONE